# A novel approach for explicit song lyrics detection using machine and deep ensemble learning models

Xiaoyuan Chen[1], Turki Aljrees[2], Muhammad Umer[3], Hanen Karamti[4], Saba Tahir[3], Nihal Abuzinadah[5], Khaled Alnowaiser[6], Ala' Abdulmajid Eshmawi[7], Abdullah Mohamed[8] and Imran Ashraf[9]

[1] Huzhou Key Laboratory of Green Energy Materials and Battery Cascade Utilization, School of Intelligent Manufacturing, Huzhou College, Huzhou, China
[2] Department College of Computer Science and Engineering, University of Hafr Al-Batin, Hafr Al-Batin, Saudi Arabia
[3] Department of Computer Science & Information Technology, The Islamia University of Bahawalpur, Bahawalpur, Pakistan
[4] Department of Computer Sciences, College of Computer and Information Sciences, Princess Nourah bint Abdulrahman University, Riyadh, Saudi Arabia
[5] Faculty of Computer Science and Information Technology, King Abdulaziz University, Jeddah, Saudi Arabia
[6] Department of Computer Engineering, College of Computer Engineering and Sciences, Prince Sattam Bin Abdulaziz University, Al-Kharj, Saudi Arabia
[7] Department of Cybersecurity, College of Computer Science and Engineering, University of Jeddah, Jeddah, Saudi Arabia
[8] Research Centre, Research Centre, Future University in Egypt, New Cairo, Egypt, New Cairo, Egypt
[9] Information and Communication Engineering, Yeungnam University, Gyeongsan si, South Korea



Corresponding authors
Turki Aljrees, tajrees@uhb.edu.sa
Imran Ashraf,
imranashraf@ynu.ac.kr

## ABSTRACT

The content of music is not always suitable for all ages. Industries that manage music content are looking for ways to help adults determine what is appropriate for children. Lyrics of songs have become increasingly inappropriate for kids and can negatively impact their mental development. However, it is difficult to filter explicit musical content because it is mostly done manually, which is time-consuming and prone to errors. Existing approaches lack the desired accuracy and are complex. This study suggests using a combination of machine learning and deep learning models to automatically screen song lyrics in this regard. The proposed model, called ELSTM-VC, combines extra tree classifier and long short-term memory and its performance is compared to other models. The ELSTM-VC can detect explicit content in English lyrics and can be useful for the music industry. The study used a dataset of 100 songs from Spotify for training, and the results show that the proposed approach effectively detects explicit lyrics. It can censor offensive content for children with a 96% accuracy. The performance of the proposed approach is better than existing approaches including machine learning models and encoding-decoding models.

## INTRODUCTION

In recent years, music has become one of the most prevalent forms of entertainment, recreation, and information. The digital music collection is growing rapidly, as artists express themselves through lyrics. However, some lyrics contain content that is not suitable for young listeners, such as references to sexual, violent, or drug-related themes. This can have negative effects on young people (*Arnett, 1991*; *Ballard & Coates, 1995*). For the past 40 years, various initiatives and measures have been taken to address this issue and prevent young people from being exposed to inappropriate musical content. Organizations in the music industry, such as the British Phonographic Industry (BPI) and the Recording Industry Association of America (RIAA), have worked to inform parents about the potential risks of certain lyrics and their potential effects on young people (*Qamar Bhatti et al., 2018*). With the increasing use of technology, particularly mobile devices, internet usage is also on the rise. According to research conducted by *The Guardian (2020)*, most children own a mobile phone by the age of seven. These devices have become an integral part of the lives of young people. With the wide proliferation of mobile phones, screening music content has become even more important for youngsters.

Protecting children from harmful music content is a big challenge for parents. In the 1990s, the music recording industry in the United States introduced parental advisory labels for albums with lyrics that contain references to sex and violence (*RIAA, 2021*). These labels are intended to alert parents that the content of the album may not be suitable for children. However, the use of these labels is voluntary and not all albums contain them, so it is not always clear which music is appropriate for young listeners. It is important for parents to be aware of the potential risks and to monitor their child's music consumption (*Cole, 2010*).

Censoring explicit content in songs is a complex task that raises several issues. One issue is that the results of screening can vary depending on the organization or individuals responsible for the judgment. In the past, there have been many controversies over the assessment of certain songs for broadcast (*Myong, 2020*). Another issue is that determining whether a song is harmful or suitable requires time and effort. Some music that might be considered inappropriate for broadcast is expressed metaphorically, making it difficult to identify without a thorough review of the lyrics. To address these challenges, it would be useful to have a way to automatically detect inappropriate content in music.

Primarily two ways exist for the automatic detection of inappropriate content in music. The first is by using a profanity dictionary. This method involves comparing the lyrics of a song to a list of words that are considered offensive or inappropriate. If any of these words are found in the lyrics, the song is classified as inappropriate. However, there is not a universal profanity dictionary, so the results of this method may vary between different companies (*Bergelid, 2018*). Additionally, if a word is not included in the profanity dictionary, the content will not be flagged as inappropriate. This approach also requires regular maintenance to keep the profanity dictionary up-to-date with new offensive language. Another way to automatically detect inappropriate content in music is through the use of machine learning (ML). ML has been used in various ways to filter inappropriate

content, particularly in the context of online user-generated content (*Xiang et al., 2012*; *Chen et al., 2012*). However, only a few attempts are made to classify inappropriate music lyrics using a machine learning model.

Natural language processing (NLP) and ML are closely related fields and are often used together to improve the capabilities of NLP models. NLP deals with the interaction between computers and human language, while ML provides a set of algorithms and techniques for training models to make predictions or decisions (*Mahedero et al., 2005*). NLP tasks such as sentiment analysis, language translation, and text summarization can benefit from ML techniques (*Joachims, 1998*). For example, a machine learning model can be trained on a dataset of labeled text to classify sentiment as positive or negative or to automatically summarize a long document.

Music lyrics are often created to attract more fans and followers, particularly among young people. These lyrics can have a significant impact on the minds of young people. In recent years, lyrics have become increasingly violent and sexual. However, current systems for filtering explicit content in music lyrics are not effective. To address this issue, a new framework is needed that uses machine learning, ensemble learning, and deep learning to classify explicit content in music lyrics. One of the challenges of this task is that song lyrics are written in a poetic notation, which makes it difficult for NLP techniques to analyze them. This study aims at developing an automatic system for filtering explicit content in music lyrics. The key contributions of this study are as follows:

- A framework is proposed that utilizes ELSTM-VC an ensemble of machine learning models extra tree classifier (ETC) and deep learning model long short-term memory (LSTM). The objective is the automatic screening of song lyrics into explicit and non-explicit content.

- Experiments are performed to evaluate the efficacy of the proposed model in comparison to other machine learning and deep learning models. Many machine learning models are employed for this purpose including logistic regression (LR), Bernoulli Naive Bayes (BNB), random forest (RF), stochastic gradient descent (SGD), decision tree (DT), AdaBoost (AB), gradient boosting machine (GBM) and ETC. In addition, LSTM deep learning model and a voting classifier are also tested which combines LR and SGD.

- Performance is evaluated using accuracy, precision, F1-score, and recall. Furthermore, k-fold cross-validation and performance comparison with several bidirectional encoder representations from Transformers (BERT) is also carried out. Empirical results prove that the ELSTM-VC model outperforms other models and is an applicable contribution to the music industry.

The rest of the article is structured as follows. 'Related Work' provides a review of related research in this field. 'Materials and Methods' presents the methodology and detailed explanation of the models used in the experiment. Results and analysis of the experiment are given in 'Results and Discussion'. 'Conclusion' discusses the implications of the research and potential areas for future works.

## RELATED WORK

A literature review plays an important role in advancing research by providing a historical perspective on the specific research area. This literature review focuses on the detection of explicit content in song lyrics. The aim is to examine how previous researchers have approached the detection of explicit content in different types of media sources.

The study of using NLP methods to identify offensive content, particularly on social media, is a widely investigated research area. For instance, *Davidson et al. (2017)* examined the detection of hate speech, which is defined as language that is meant to express hatred towards a specific group or to demean, humiliate, or insult the members of that group. To classify the tweets into three categories the authors used the LR classifier. The system performed well overall, but had difficulty correctly identifying hate speech; 40% of hate speech is misclassified. *Corazza et al. (2020)* recently suggested a modular neural network design for detecting hate speech that can be applied to multiple languages such as English, Italian, and German. Their approach incorporates a variety of features, including social media network features, text-based features, FASTTEXT word embeddings, and emotion lexica. To classify the toxic comments *Carta et al. (2019)* proposed a supervised multi-class multi-label word embeddings approach, such as verbal bullying and personal attacks. The dataset is obtained from Wikipedia's Talk page, and the results show that using sets of word embeddings improves performance compared to the traditional bag-of-words (BOW) model.

A few recent studies have focused on using NLP to automatically detect explicit lyrics. *Chin et al. (2018)* tackled the problem of detecting offensive content in Korean songs using a *corpus* of 27,695 Korean song lyrics, 1,024 of which are labeled as explicit (3.7% of the total). The authors established a baseline for detecting explicit content by utilizing a profanity language dictionary from Namu-wiki, a Korean wiki, and achieved a macro F1-score of 0.61 (0.88 for the weighted F1). The authors tested various machine learning classification algorithms, such as AdaBoost and Bagging, and found that Bagging performed the best, with an F1-score of 0.78 and a weighted F1 of 0.96.

*Kim & Mun (2019)* aimed to identify explicit content in Korean song lyrics by utilizing a *corpus* of 70,077 lyrics, with 10.7% labeled as explicit. They tested a lexicon-based filtering approach, using a dictionary of explicit words generated automatically. They employed a Hierarchical attention networks (HAN) model and a recurrent neural network (RNN)-based model for processing words sequentially and hierarchically. The results reveal that the combination of HAN and vector representations of explicit words produce from the lexicon performs the best, achieving an F1-score of 0.805. In the study by *Rospocher (2020)*, the effectiveness of FastText word embedding and a classifier is tested on a large dataset of English song lyrics. The dataset consisted of 807,707 songs, with 7.74% of them labeled as explicit. The results show that FastText performs better than other methods such as LR and majority vote, proposing that using more advanced classifiers could improve performance even further.

*Fell et al. (2019a)* conducted a study that evaluated machine and deep learning techniques for classifying explicit lyrics, using the largest dataset to date with 179,391

English song lyrics, 17,808 of which are marked as explicit. The study compared methods including dictionary-based methods, an LR classifier, a BERT language model, and textual Deconvolution Saliency. The results show that while textual Deconvolution Saliency performs the best with an F1-score of 0.796, deep learning models did not outperform the other simpler methods. *Fell et al. (2019b)* conducted a follow-up study where they applied an LR classifier with term frequency-inverse document frequency (TF-IDF) BoW vector representations to a dataset of 438,000 song lyrics, achieving an F1-score of 0.773.

*Lin, Tseng & Fuh (2003)* proposed an explicit content detection algorithm based on a support vector machine (SVM) for classification. SVM is a technique that uses statistical learning theory to analyze, predict, and identify explicit content in images. The proposed model had a low accuracy value for explicit content detection. The study by *Kia et al. (2014)* focused on identifying pornographic images using a combination of techniques. The authors proposed using two specific features related to the skin region and Fourier descriptors. These features are found to improve the accuracy of identifying explicit images when compared to traditional methods. They also employed a combination of a multilayer perceptron and Neuro Fuzzy using fuzzy integral information. The system is tested and found to have a precision of 93% for true positives and 8% for false positives on the training dataset, and 87% and 5.5% on the testing dataset.

In a nutshell, the literature review shows that the detection of explicit content in song lyrics using NLP techniques is an active area of research. There are multiple applications of machine learning, deep learning, and ensemble learning models in terms of cyber security (*Ashraf et al., 2022*; *Majeed et al., 2021*), text classification (*Umer et al., 2023*; *Karim et al., 2022*), drinking water quality prediction (*Juna et al., 2022*; *Madni et al., 2023*), and digital healthcare problems (*Hafeez et al., 2022*; *Ahmed et al., 2023*). For text classification, various approaches have been proposed, including using a lexicon-based filtering approach, machine learning classification algorithms, and deep learning methods such as RNNs and CNNs. These studies have been conducted on different languages and datasets, and have achieved varying levels of performance. Some studies suggest that further improvement may be achieved by using more advanced machine learning classifiers.

## MATERIAL AND METHODS

This section describes the architecture used to predict explicit content in music lyrics. The study employs a variety of tools, methods, and techniques to detect explicit content in song lyrics. The pipeline of the adopted methodology is shown in Fig. 1. The first step is the collection of a dataset, followed by preprocessing of the data. Finally, the classifiers used in the experiment are discussed.

### Dataset

This research utilizes a dataset obtained from GitHub, a widely used platform for researchers. The dataset, named 'Spotify Song Lyrics Analysis' is provided in a CSV file and includes the lyrics of Billboard Hot 100 song singles from 1965 to 2015 (*Zhao, 2018*). The goal of the research is to identify songs with explicit lyrics using various machine learning, ensemble learning, and deep learning techniques. The research used a dataset with 30

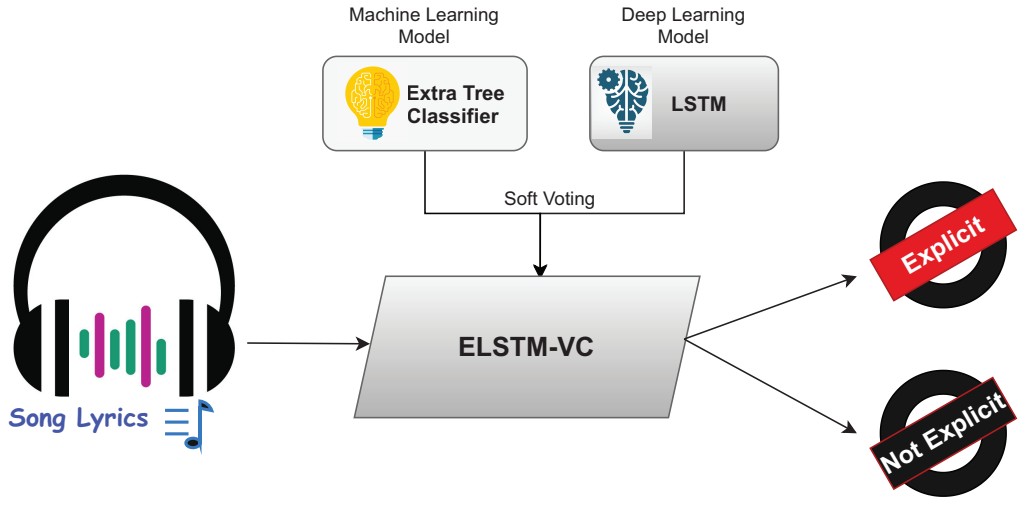

**Figure 1 Pipeline of the proposed framework.**

columns, including 'uniq ratio', 'rank', 'instrumentalness', 'artist base', 'song', 'song clean', 'lyrics', 'acousticness', 'energy', 'key', 'loudness', 'artist clean', 'popularity', 'speechiness', 'danceability', 'tempo', 'explicit', 'artist featured', 'valence', 'year', 'duration min', 'mode', 'time signature', 'num words', 'liveness', 'num uniq words', 'decade', 'words per sec', 'artist all' and 'release date'. The dataset's target variable is 'Explicit' which is binary, with a value of '1' representing explicit lyrics and '0' representing non-explicit lyrics. An overview of the dataset, including a description of each attribute, is provided in Table 1.

### Data preprocessing

This research used preprocessing techniques to convert raw data into a format that is easier to analyze and improves the efficiency of the learning process. The preprocessing is done using the Ski-kit learn library and the NLP toolkit (NLTK) in Python (*Han, Kamber & Pei, 2011*; *Vijayarani & Janani, 2016*). Some of the specific preprocessing steps that are applied to the dataset include tokenization, removal of stop words, removal of punctuation, removal of numbers, correction of spelling, and stemming. These steps are used to clean and prepare the data for analysis.

## Brief description of learning classifiers for explicit lyrics detection

This study employed various machine learning techniques including tree-based, regression-based, and ensemble models to detect explicit song lyrics. The specific models utilized are LR, GBM, BNB, RF, DT, ETC, SGD, LSTM, and two voting classifiers ETC +LSTM, RF-LSTM, BNB-LSTM, and LR+SGD. A list of hyperparameters and their concerning values used for experiments are shown in Table 2 while a brief description of these algorithms is given in Table 3.

## Deep learning classifiers for explicit lyrics detection

This study makes use of long short-term memory as a baseline deep learning model for detecting explicit song lyrics.

**Table 1 Main characteristics of song lyrics dataset used in this study.**

| Sources | LyricsWiki, Spotify |
|---|---|
| Years coverage | 1950–2019 |
| Language | English |
| Number of lyrics | 807,707 |
| Number of explicit lyrics | 62,549 |
| % of explicit lyrics | 7.74% |

**Table 2 Hyperparameter values of all learning models used for explicit song lyrics detection.**

| Model | Parameters |
|---|---|
| RF | Number of trees = 200, Maximum depth = 30, Random state = 52 |
| DT | Number of trees = 200, Maximum depth = 30, Random state = 52 |
| LR | Penalty = 'l2', Solver = 'lbfgs' |
| GBM | Number of trees = 200, Maximum depth = 30, Random state = 52, Learning rate = 0.1 |
| ETC | Number of trees = 200, Maximum depth = 30, Random state = 52 |
| BNB | Alpha = 1.0, Binarize = 0.0 |
| SGD | Penalty = 'l2', Loss = 'log' |
| LSTM | LSTM (390 units, return_sequences = True), LSTM (260 units, return_sequences = True), Dropout (0.2), LSTM (190 units, return_sequences = True), Dropout (0.2), LSTM (60 units, return_sequences = True), Dropout (0.2), Dense (2 neurons), optimizer = 'adam' |

**Table 3 Brief description of machine learning models.**

| Model | Description |
|---|---|
| Decision tree | A DT is a model with a tree-like structure that can perform both classification and regression tasks. It has a flowchart-like appearance where a feature or attribute is represented by an internal node, a decision rule by a branch, and the outcome by a leaf node (*Hand, 2013*). The root node is the top node in the tree. The data is split into smaller subsets based on the values of the features and the algorithm selects the best feature and threshold value to separate the data according to the target variable. This process continues on each subset until a pre-defined stop criterion, such as maximum tree depth or minimum samples per leaf, is reached. |
| AdaBoost classifier | AdaBoost (Adaptive Boosting) is a widely used ensemble learning technique for both classification and regression problems. It is a type of boosting algorithm that combines multiple simple models (referred to as weak learners) such as decision stumps or single split decision trees, to create a stronger overall classifier (*Freund & Schapire, 1997*). The algorithm starts by training a weak learner on the training data and making predictions. The misclassified samples are then given higher weights and a new weak learner is trained on the re-weighted data. The predictions of the first weak learner and the predictions of the new weak learner are then combined to make a final prediction. This process is repeated multiple times to create a sequence of weak learners, where each learner aims to correct the errors of the previous learners. |
| Logistic regression | LR is a machine learning algorithm used for classification problems. It is a form of supervised learning which uses a logistic function to model a binary dependent variable with two possible outcomes (*e.g.*, true/false, pass/fail, 0/1) (*Mitchell, 2006*). The logistic function, also known as the sigmoid function, maps the input features to a value between 0 and 1, which represents the probability of the positive class. Logistic regression makes predictions by thresholding the probability at a certain value (*e.g.*, 0.5) and classifying the samples as positive or negative. Additionally, logistic regression can handle multiple classes by using a one-*vs*-all or softmax approach, which creates multiple binary classifiers for each class. |

(Continued)

| Table 3 (continued) | |
|---|---|
| **Model** | **Description** |
| Stochastic gradient descent | SGD is an optimization algorithm that is commonly used to train machine learning models, particularly in cases where the dataset is large and the model is complex. It is an iterative method that is used to minimize the objective function, which is typically the cost function of the model. In each iteration of the algorithm, a small randomly selected subset of the data called a mini-batch is used to calculate the gradient of the cost function concerning the parameters of the model. The parameters are then updated in the opposite direction of the gradient, using a learning rate that determines the step size. This process is repeated until the cost function converges or reaches a stopping criterion (*Zadrozny & Elkan, 2002*). |
| Random forest | RF is a type of ensemble learning method for classification and regression. It is a collection of decision trees, where each tree is trained on a different subset of the data and the final prediction is made by averaging or voting the predictions of the individual trees. The algorithm creates multiple decision trees by randomly selecting a subset of the features and a subset of the training examples to grow each tree. This process is repeated multiple times to create a forest of decision trees. The idea behind this approach is to reduce the variance of the model by averaging multiple trees, each of which may have a high bias but a low variance. The randomness in the selection of features and examples used to train each tree also helps to reduce overfitting, which is a common problem in decision trees. The final prediction is made by averaging the predictions of the individual trees in case of regression and by taking the mode of the class predictions in case of classification (*Gregorutti, Michel & Saint-Pierre, 2017*). |
| Gradient boosting machine | GBM is an ensemble learning method for classification and regression problems. It is a powerful and popular algorithm that is used to improve the performance of decision trees by combining multiple weak learners. The algorithm starts by training a simple decision tree on the training data and making predictions. The residual errors (difference between the predictions and the true values) are then calculated and a new tree is trained to predict these residual errors. The predictions of the first tree and the predictions of the new tree are then combined to make a final prediction. This process is repeated multiple times to create a sequence of decision trees, where each tree aims to correct the errors of the previous trees (*Friedman, 2001*). |
| Extra tree classifier | ETC, also known as extremely randomized trees classifier, is an ensemble learning method for classification problems. It is similar to random forest classifier, but it is more random and results in a higher degree of decorrelation between the trees. Like random forest, the extra trees classifier builds multiple decision trees by randomly selecting a subset of the features and a subset of the training examples to grow each tree. However, instead of using the traditional method of finding the best-split point, it randomly selects the split point for each feature at each internal node. This results in a higher degree of randomness and decorrelation between the trees (*Breiman, 2001*). The final prediction is made by averaging the predictions of the individual trees. The ETC is considered to be a more random and less correlated version of the RF classifier, which can lead to better generalization performance on unseen data. The ETC is also more efficient and faster to train than the random forest classifier. It is considered to be a good choice for large datasets and high-dimensional feature spaces. However, it is less interpretable than random forest classifier and it doesn't provide feature importance. |
| Bernoulli Naive Bayes | BNB is a probabilistic algorithm that is primarily used for binary classification problems. It is a variation of the Naive Bayes algorithm, which is a simple and efficient method based on Bayes' theorem and the assumption of independence between the features. Bernoulli Naive Bayes assumes that the features are binary (*i.e.*, they can only take the values 0 or 1) and that they are conditionally independent given the class variable. It uses the Bernoulli distribution to model the probability of the features being 1 given the class. It then uses Bayes' theorem to calculate the probability of a new sample belonging to each class and make a prediction based on the class with the highest probability. It is called "Naive" because it assumes that all the features are independent. This assumption is often not true, but in practice, the algorithm often still performs well. Additionally, Bernoulli NB is less complex and computationally cheaper than multinomial Naive Bayes which is used for discrete data (*Kim et al., 2006*). |
| Voting classifier | Mostly voting classifier is used for classification problems because it allows the combination of two or more learning models to train on the dataset (*Lam & Suen, 1997*; *Ruta & Gabrys, 2005*). In the voting classifier for the sample data point, each model predicts a result. This result is taken as a 'vote' in favor of the class that model predicted. Once each model gives the outcome, the final prediction is based on the majority of the vote for the specific class. In most of the voting classifiers, the number of classifiers depends on the need, and the target label with the most votes is considered final. In this study, two voting classifiers are used. One voting classifier is the ensemble of the two ML models. These models are LR and SGD and the other is an ensemble of machine learning and deep learning model that are ETC and LSTM. |

### Long short-term memory

The LSTM model is a deep learning system that is specifically designed to handle text classification tasks effectively. It can handle sequential data, and it also preserves information from previous inputs through its three gates: input gate ($ik$), output gate ($Ok$), and forget gate ($fk$) (*Gers, Schmidhuber & Cummins, 1999*; *Bengio, Simard & Frasconi, 1994*; *Hochreiter & Schmidhuber, 1997*). These gates determine which information is relevant to the classification and which can be discarded, based on the dropout value. Any important information for the prediction is stored in the cell memory block. There are different variations of LSTM, but the one used in this case is as follows

$$f_t = \alpha(W - [h_{t-1}; w_t + b_f]) \tag{1}$$

where $\sigma$ is the sigmoid function, $W_f$ shows the weight of weight vector of input, $ht-1$ is the forecast vector from the previous period, $x_t$ is the new input vector $b_f$ is the bias of function $f$, and $b_f$ bias coefficient is the common function of all the machine learning functions which can be calculated either beforehand or it can be calculated during the training process. $b_f$ is also used to calibrate the model. Bias calibrations are quite helpful for the detection of explicit lyrics.

The next step of LSTM is to analyze the new data to find whether it is useful or not. If data is useful then it is added to the model memory. Nonetheless, the significance of the new data needed to be scaled, just like in the 'forget' function. So, we need to calculate the 'include' function, deployed in the corresponding layer *i.e.*,

$$i_t = \alpha(W - [h_{t-1}; w_t + b_i]) \tag{2}$$

To find the new candidate vector (vector $c_t$) this can be added to the neural cell state which can be represented as follows

$$c_t = i_t \odot g_t + f_t \odot c_{t-1} \tag{3}$$

If we determine the information from the above-mentioned three equations we can find the new cell state $c_t$. The new cell state is the outcome of multiplying the old state, $Ct-1$, by $ft$, forgetting anything which seems to be useless in the first step and then adding $i_t \times c_t$ which results in the new candidate values scaled by how much decided to update each state value. In the 'update' layer the new cell state is calculated as

$$c_t = f_t \times c_{t-1} + c_t \times i_t. \tag{4}$$

The other equations of LSTM are as follows

$$h_t = o_t \odot tanh(c_t) \tag{5}$$
$$o_t = \alpha(W - [h_{t-1}; w_t + b_o]) \tag{6}$$

where $\odot$ stands for element-wise multiplication, $\alpha$ is sigmoid function, $W_i, W_f, b_i, W_o, b_f, b_o$ are the input, output and forget gate parameters.

After computing the hidden vector of every position, the LSTM model regards the last hidden vector as the sentence representation. After that, it has been fed to a layer with an

output length of the class number, and the softmax layer is added to generate the output of classifying sentence probability as positive, neutral, or negative. The softmax function is computed as follows, where C is the count of sentiment classes.

$$softmax_i = exp(x_i)/\sigma i_t = 1^C exp(x_{it}). \tag{7}$$

Like other layers of the neural network, the dense layer is made up of a deeply connected neural network. Every neuron in this layer takes the output values from the past layer's neurons. The dense layer on the input features map performs vector-matrix multiplication. The input values of the dense layer are the parameters that can be trained using the backpropagation techniques. The main function of this layer is to transform the shape of the vector based on dimensions.

$$Output = activation(dot(input, kernel) + bias) \tag{8}$$

where *dot* shows the dot product of input values and its corresponding weighted values, and *bias* is the biased value used in the ML for the model optimization.

The dropout rate is a regulation method that is used in this study for explicit lyrics detection. The proposed of using of the dropout layer is to reduce the model's complexity and to avoid the overfitting phenomena as well. This is used by randomly deactivating a specific number of neurons linked with the layers using the 'P' probability from the Bernoulli distributions. In the training of the system, the pass of the feedforward neural network cannot rely on the performance of the specific activation. In the end, the neural network will train on diverse, irrelevant features.

## Proposed methodology

The proposed methodology in this study combines the use of an ensemble of a deep neural network model and a machine learning model to detect explicit song lyrics. The use of deep neural networks has been gaining popularity among researchers in recent years as they have been shown to increase accuracy compared to traditional classifiers. Thus, a deep neural network, specifically an ELSTM-VC, which combines ETC and LSTM using soft voting, is used in this study. In the soft voting criteria, the final prediction is based on the class with the highest probability, as illustrated in Fig. 2.

Algorithm 1 presents the steps for the proposed ELSTM-VC, which combines ETC and LSTM for explicit song lyrics classification. The algorithm uses ETC and LSTM as two methods and 'explicit' and 'not-explicit' as the two classes, and the prediction is made using the following equation:

$$H_{Prob} = argmax\{Expl_{prob}, NExp_{prob}\}. \tag{9}$$

The proposed model, ELSTM-VC uses a combination of ETC and LSTM models to classify explicit song lyrics. The algorithm calculates the highest predicted probability Hprob using the argmax function. The joint probability of the explicit class and the Not-explicit class is represented by Exprob and NExprob, respectively, and are computed using the following equation

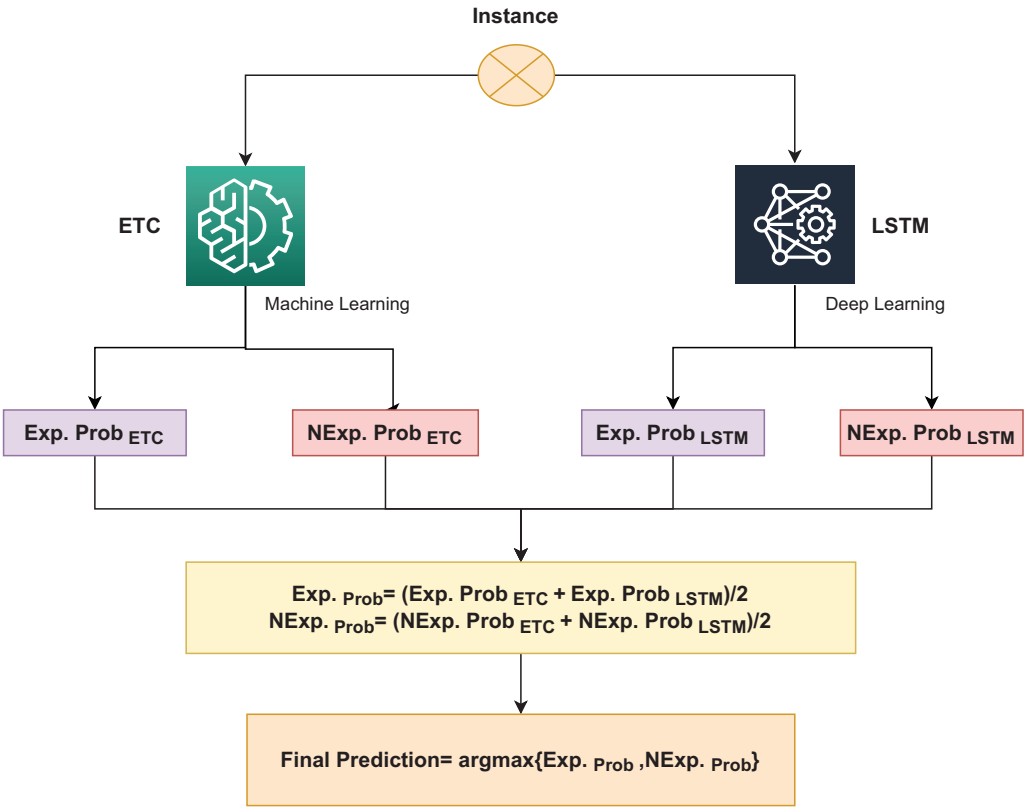

**Figure 2  Architecture of the proposed ELSTM-VC.**

---

**Algorithm 1  Ensembling of ETC and LSTM (ELSTM-VC).**

**Input:** input data $(x, y)_{i=1}^N$

$T_{ETC}$ = Trained_ ETC

$T_{LSTM}$ = Trained_ LSTM

1: **for** $i = 1$ to $T$ **do**

2:    **if** $T_{ETC} \neq 0$ & $T_{CNN} \neq 0$ & $training\_set \neq 0$ **then**

3:        $ProbLSTM - Exp = T_{LSTM}.probability(Exp - class)$

4:        $ProbLSTM - NExp = T_{LSTM}.probability(NExp - class)$

5:        $ProbETC - Explicit = T_{ETC}.probability(Exp - class)$

6:        $ProbETC - NExp = T_{ETC}.probibility(NExp - class)$

7:        Decision function $= max\left(\dfrac{1}{N_{classifier}}\sum_{classifier}\left(Avg_{(ProbLSTM-Exp, ProbETC-Exp)},\right.\right.$
            $\left.\left. Avg_{(ProbLSTM-NExp, ProbETC-NExp)}\right)\right)$

8:    **end if**

9:    Return final label $\hat{p}$

10: **end for**

---

$$Exp_{prob} = \frac{ExpProb_{ETC} + ExpProb_{LSTM}}{2} \tag{10}$$

$$NExp_{prob} = \frac{NExpProb_{ETC} + NExpProb_{LSTM}}{2} \tag{11}$$

where $ExpProb_{ETC}$, and $ExpProb_{LSTM}$ are the probability for the explicit class by ETC and LSTM respectively, on the other hand, $NExpProb_{ETC}$, and $NExpProb_{LSTM}$ are the non-explicit class probability score for ETC and LSTM. For the detailed description of the voting classifier, the values are obtained from the data sample. ETC and LSTM give the probabilities for the sample are

$ExpProb_{ETC} = 0.7$
$NExpProb_{ETC} = 0.5$
$ExpProb_{LSTM} = 0.6$
$NExpProb_{LSTM} = 0.4$
$Exp_{Prob}$ and $NExp_{Prob}$ is calculated as

$$Exp_{Prob} = \frac{0.7 + 0.6}{2} \tag{12}$$

$$NExp_{Prob} = \frac{0.5 + 0.4}{2} \tag{13}$$

Then probability values are passed to the argmax function, which returns the highest probability class.

$$H_{Prob} = argmax(0.65, 0.45) \tag{14}$$

In this case, the highest predicted probability is for the explicit class, so the final prediction will be for the explicit class. The architecture of the proposed approach is illustrated in the above figure. The research also uses various machine learning models that use Word2vec as a feature representation technique as a baseline for comparison with the proposed ELSTM-VC model. The dataset is divided into two parts with a 20% and 80% ratio for testing and training, respectively. The training set is used to train all machine learning models and the proposed ELSTM-VC. The performance of all models is then evaluated using the test data.

## Evaluation parameters

This study presents a deep learning method that utilizes LSTM and word embedding to classify lyrics as explicit or non-explicit. The dataset is divided into two categories: normal and explicit. The proposed approach is evaluated using several evaluation metrics, including accuracy, precision, recall, F1-score, and specificity, to assess its effectiveness (*Umer et al., 2022*; *Manzoor et al., 2021*).

$$Accuracy = \frac{TP + TN}{TP + TN + FP + FN} \tag{15}$$

$$Precision = \frac{TP}{TP + FP} \tag{16}$$

$$Recall = \frac{TP}{TP + FN} \tag{17}$$

$$F1 - score = 2 \times \frac{precision \times recall}{precision + recall} \tag{18}$$

where TP, TN, FP, and FN are true positive, true negative, false positive, and false negative, respectively, and extracted from the confusion matrix.

## RESULTS AND DISCUSSION

In this research, an ensemble method combining a deep learning model and a machine learning model is proposed. The performance of this ensemble is compared to other base models in terms of efficiency. The classifiers are evaluated using performance metrics such as accuracy, precision, recall, and F1-score. In this research, for the detection of the explicit song lyrics tree-based and regression-based models are used individually and in an ensemble, and deep learning models are also used individually and in an ensemble with machine learning models are used.

### Experimental setup

The experiments are conducted using Python programming language with supporting libraries like Sklearn, Tensorflow, Keras, Pandas, Numpy, and Matplotlib. The code runs on the Anaconda Jupyter Notebook environment. The main reason for using this environment is that it automatically downloads and loads required libraries. Model training is performed on a high-performance 2 GB Dell PowerEdge T430 server with a graphical processing unit (GPU) 2X Intel Xeon 8 cores and 32 GB of DDR4 RAM. The entire process takes approximately 20 min to complete.

### Results of machine and deep learning models

Table 4 shows the results of employed models regarding accuracy, precision, recall, and F1-score. Among the machine learning models, RF and ETC achieve the highest accuracy of 91% and perform well in detecting explicit song lyrics.

Experimental results in Table 4 show that the proposed ELSTM-VC and LSTM models alone achieved a precision of 100%, while the ETC model alone has a precision of 92.38%. The reason for making an ensemble of ETC and LSTM is that they both have the highest accuracy as individual models. In the employed machine learning models, ETC has the best results while deep LSTM has the best results among deep learning models. These models are joined as an ensemble and good accuracy is expected. For further confirmation, other ensemble models are also designed involving the second and third best-performing machine learning models with LSTM but the results are not higher than the ELSTM-VC model. The highest recall value is achieved by the ETC and RF models; both have 92% recall. The LSTM model has the second-highest recall value which is 89%. The LSTM model also has the highest F1-score of 91%, while the RF and ETC models also performed well with an F1-score of 90%. However, the best accuracy and precision are obtained by the

**Table 4 Comparison of machine learning and deep learning models in terms of precision, recall, F1-score.**

| Models | Accuracy | Precision | Recall | F1-score |
|---|---|---|---|---|
| DT | 75.34% | 80.25% | 75.37% | 77.41% |
| AB | 75.19% | 80.45% | 75.67% | 77.54% |
| LR | 83.33% | 69.49% | 83.72% | 76.58% |
| SGD | 75.37% | 80.29% | 75.31% | 77.55% |
| RF | 91.17% | 92.85% | 92.37% | 90.35% |
| GBM | 75.36% | 80.82% | 75.74% | 77.69% |
| ETC | 91.70% | 92.38% | 92.19% | 90.52% |
| BNB | 83.45% | 69.65% | 83.21% | 76.44% |
| CNN | 88.34% | 82.68% | 85.37% | 83.62% |
| LSTM | 92.28% | 100.00% | 89.17% | 91.31% |
| Voting classifier (LR+SGD) | 83.74% | 69.59% | 83.47% | 76.34% |
| BLSTM-VC | 91.27% | 93.29% | 94.55% | 94.27% |
| RLSTM-VC | 93.19% | 95.68% | 96.29% | 95.98% |
| **ELSTM-VC** | **96.92%** | **100.00%** | **92.97%** | **96.79%** |

Note:
  The proposed model is indicated in bold.

proposed model which achieves a 96.92% accuracy which is far better than all other models employed in this study.

Among tree-based algorithms, the decision tree has the worst performance with an accuracy of 75% as shown in Table 4. Decision trees often struggle to handle diverse features and tend to overfit. However, advanced tree-based models like ETC and RF show improved stability and performance. This is why the ETC and RF classifiers perform well in detecting explicit music lyrics.

The deep learning models use a technique called word embedding. LSTM using word embedding has higher results than other machine learning classifiers. This suggests that word embedding has a significant impact on detecting explicit lyrics in songs. The proposed ELSTM-VC outperforms all other models with 96% accuracy, 100% precision, 95% recall, and 97% F1-score. The proposed approach which is an ensemble of a machine learning and deep learning model performs better using Word2vec features which are the highest of all classifiers. A comparison of the results of all machine learning models and the deep learning model is illustrated in Fig. 3.

## Results of K-fold cross-validation

To verify the effectiveness of the proposed model, this research work makes use of k-fold cross-validation. Table 5 provides the results of the 10-fold cross-validation. Cross-validation results reveal that the proposed ensemble model provides an average accuracy score of 96.95% while the average scores for precision, recall, and F1-score are 99.88%, 92.74%, and 96.75%, respectively.

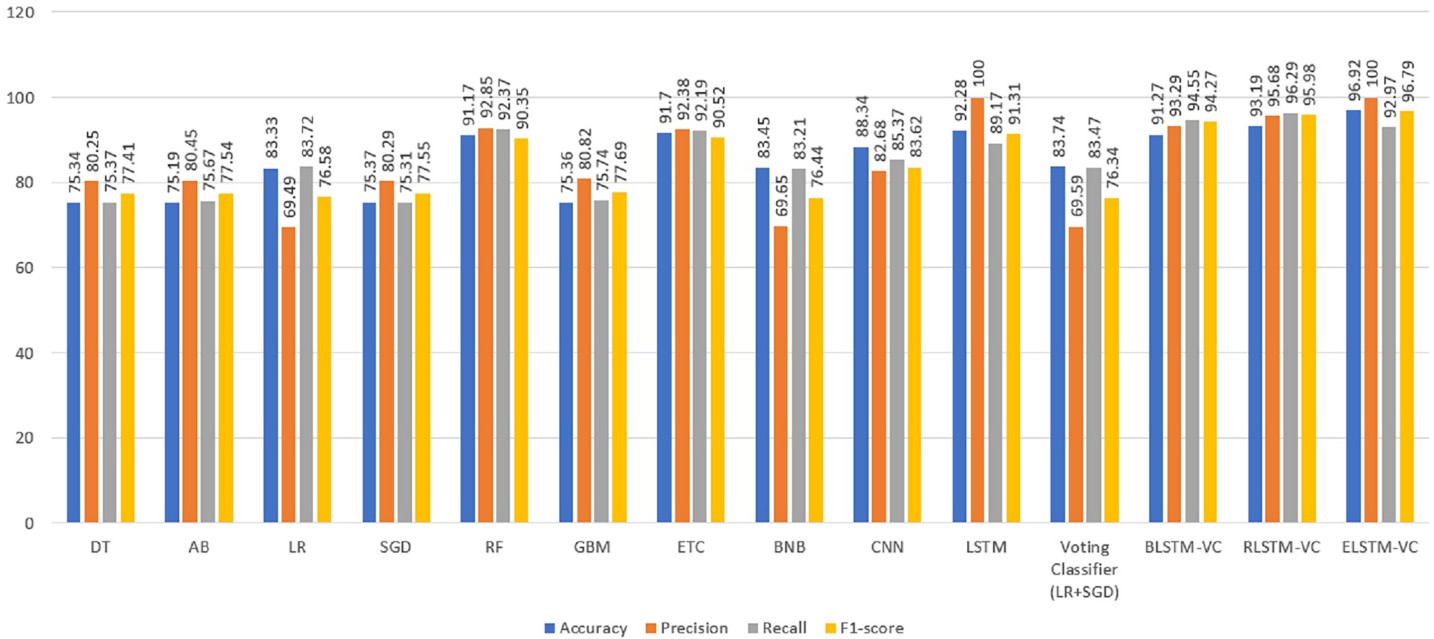

**Figure 3** Performance comparison of classifiers in terms of accuracy, precision, recall, and F1 score.

**Table 5 Results of 10-fold cross-validation for the proposed approach in percentage (%).**

| Fold number | Accuracy | Precision | Recall | F1-score |
|---|---|---|---|---|
| Fold-1 | 95.25 | 99.51 | 92.44 | 96.55 |
| Fold-2 | 95.43 | 99.64 | 92.55 | 96.64 |
| Fold-3 | 95.63 | 99.73 | 92.63 | 96.79 |
| Fold-4 | 95.84 | 99.99 | 92.98 | 96.84 |
| Fold-5 | 95.94 | 99.99 | 92.83 | 96.88 |
| Fold-6 | 95.92 | 99.99 | 92.97 | 96.81 |
| Fold-7 | 96.55 | 99.99 | 92.65 | 96.79 |
| Fold-8 | 96.74 | 100.00 | 92.73 | 96.82 |
| Fold-9 | 96.77 | 100.00 | 92.82 | 96.82 |
| Fold-10 | 96.95 | 100.00 | 92.94 | 96.99 |
| **Average** | **95.95** | **99.88** | **92.74** | **96.75** |

## Comparison with encoding-decoding models

To compare the proposed strategy with the most recent models, this study applied BERT, PHS-BERT, and BioALBERT models. Using a sizable dataset of 3.3 billion words, the success of BERT has been analyzed. BERT is trained on the 800 million words BooksCorpus of Google and the 2.5 billion word Wikipedia. BERT is unique from other conventional models since it can read simultaneously in both directions. Bidirectionality is

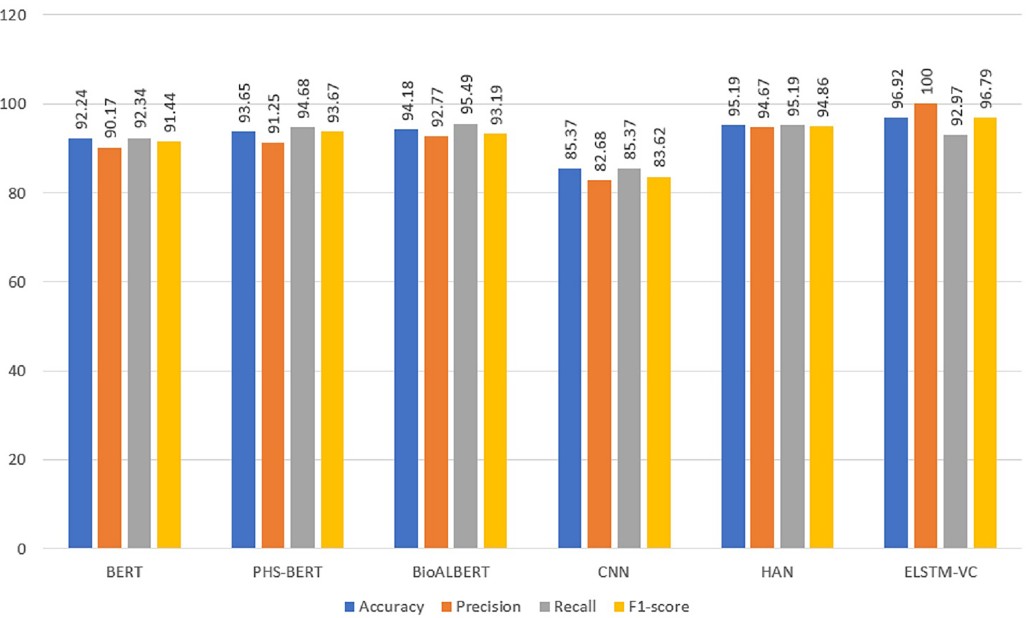

**Figure 4 Performance comparison of the proposed model with transformer-based and state-of-the-art models.**

**Table 6 Comparison with transformer-based models.**

| Model | Accuracy | Precision | Recall | F1-score |
|---|---|---|---|---|
| BERT | 92.24% | 90.17% | 92.34% | 91.44% |
| PHS-BERT | 93.65% | 91.25% | 94.68% | 93.67% |
| BioALBERT | 94.18% | 92.77% | 95.49% | 93.19% |
| CNN | 88.34% | 82.68% | 85.37% | 83.62% |
| HAN | 94.08% | 94.67% | 95.19% | 94.86% |
| **ELSTM-VC** | **96.92%** | **100.00%** | **92.97%** | **96.79%** |

Note:
The proposed model is indicated in bold.

the term for this capacity, which was made possible with the help of transformers. BERT is outperforming in different 11 NLP tasks. PHS-BERT is proposed for tasks related to public health surveillance and has proved its robustness (*Naseem et al., 2022*). BioALBERT 1.1 is a large model that is trained on biomedical *corpus* and outperforms other models on many datasets (*Naseem et al., 2021*). We also added a comparison with state-of-the-art models like convolutional neural networks (CNN) and hierarchical attention networks (HAN) from *Hameed et al. (2021)*, *Umer et al. (2021)*.

Results of the transformer-based models are given in Table 6 and in Fig. 4. It can be observed that all three models have achieved an accuracy greater than 92%. BioALBERT has attained the highest accuracy with 94.18% which is closest to the proposed model. If we compare both models in terms of complexity, it can be noticed that BioALBERT is a multilayer complex deep learning model, and is trained on millions of parameters that require high computational power. Still, the accuracy of transformer-based models is not

greater than the proposed model. Transformer-based models did not perform well because transformer models are computationally expensive and require large amounts of data to train effectively. Owing to comparatively smaller training data, the performance of transformer-based models is not good enough. The transformer model struggles to generalize and performs poorly on new examples. Contrarily, the proposed model is an ensemble-learning model and can be executed on lower-power machines. Thus the proposed model is more suitable for performing explicit song lyrics detection. The performance of CNN is not good but HAN performs second best in terms of accuracy.

## CONCLUSION

Explicit song lyrics detection is becoming increasingly important to detect references to sex, violence, and drug-related themes. Automated explicit content detection methods are needed to cope with the large use of mobile phones. This research work presents an ensemble model that utilized ETC and LSTM for detecting explicit lyrics. Experiments are performed on a large dataset employing many machine-learning models. Results reveal that the proposed model ELSTM-VC achieves a 96.0% accuracy which is the highest compared to other machine learning models. The results also indicate that the proposed model performs better than other state-of-the-art models like BERT, PHS-BERT, and BioALBERT for song explicit lyrics detection. However, it is important to note that the computational complexity of the proposed approach is higher than standalone machine learning models which will be a focus of future research. Future work will also involve analyzing important features using dimensional reduction techniques and testing other deep learning models for explicit lyrics detection.

## ABBREVIATIONS

| | |
|---|---|
| **CNN** | Convolutional Neural Network |
| **ETC** | Extra Tree Classifier |
| **ANN** | Artificial Neural Network |
| **BPI** | British Pornography Industry |
| **LR** | Logistic Regression |
| **RIAA** | Recording Industry Association of America |
| **NLP** | Natural Language Processing |
| **ML** | Machine Learning |
| **VC** | Voting Classifier |
| **BERT** | Bidirectional Encoder Representations from Transformers |
| **BoW** | Bag of Words |
| **RF** | Random Forest |
| **TF** | Term-Frequency |
| **IDF** | Inverse-document-frequency |
| **SGD** | Stochastic Gradient Descent |
| **GBM** | Gradient Boosting Machine |
| **AB** | AdaBoost |
| **BNB** | Bernoulli Naive Bayes |

| **TP** | True Positive |
|---|---|
| **FP** | False Positive |
| **FN** | False Negative |
| **TN** | True Negative |

## ACKNOWLEDGEMENTS

I am deeply grateful to all those who contributed to this article and those who played a big role in the success of this article. I would like to thank the University of Hafr Al Batin for their invaluable support and Muhammad Umer for his insights and expertise, which were instrumental in shaping the direction of this research process.

### Funding

This work was supported by the Princess Nourah bint Abdulrahman University Researchers Supporting Project number (PNURSP2023R192), Princess Nourah bint Abdulrahman University, Riyadh, Saudi Arabia. The funders had no role in study design, data collection and analysis, decision to publish, or preparation of the manuscript.

### Grant Disclosures

The following grant information was disclosed by the authors:
Princess Nourah bint Abdulrahman University Researchers: PNURSP2023R192.
Princess Nourah bint Abdulrahman University, Riyadh, Saudi Arabia.

### Competing Interests

Imran Ashraf is an Academic Editor for PeerJ Computer Science.

### Author Contributions

- Xiaoyuan Chen performed the experiments, authored or reviewed drafts of the article, and approved the final draft.
- Turki Aljrees analyzed the data, authored or reviewed drafts of the article, and approved the final draft.
- Muhammad Umer conceived and designed the experiments, analyzed the data, prepared figures and/or tables, and approved the final draft.
- Hanen Karamti conceived and designed the experiments, analyzed the data, prepared figures and/or tables, and approved the final draft.
- Saba Tahir conceived and designed the experiments, performed the computation work, authored or reviewed drafts of the article, and approved the final draft.
- Nihal Abuzinadah performed the experiments, performed the computation work, prepared figures and/or tables, and approved the final draft.
- Khaled Alnowaiser performed the experiments, performed the computation work, prepared figures and/or tables, and approved the final draft.

- Ala' Abdulmajid Eshmawi conceived and designed the experiments, performed the computation work, authored or reviewed drafts of the article, and approved the final draft.
- Abdullah Mohamed performed the experiments, analyzed the data, authored or reviewed drafts of the article, and approved the final draft.
- Imran Ashraf performed the experiments, analyzed the data, authored or reviewed drafts of the article, and approved the final draft.

## Data Availability

The data is available at GitHub and Zenodo:

https://github.com/MUmerSabir/ExplicitSongsLyrics.

MUmerSabir. (2023). MUmerSabir/ExplicitSongsLyrics: PeerJCS (MasterPeerJCS).

Zenodo. https://doi.org/10.5281/zenodo.7711818.

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
