# Peer review of "A novel approach for explicit song lyrics detection using machine and deep ensemble learning models"

_PeerJ Computer Science, doi:10.7717/peerj-cs.1469_

## Round 0.1 · original submission · Major Revisions

Dear authors,

Thank you for submitting your manuscript to PeerJ Computer Science.

We have completed the evaluation of your manuscript. The reviewers recommend reconsideration of your manuscript following major revision. I invite you to resubmit your manuscript after addressing the comments made by the reviewers. In particular:
1. Extend the abstract as indicated by reviewer 1 and 3.
2. Include the tables and figures requested by the reviewers.
3. Justify the choice of methods.
4. Include a more in-depth discussion of the results.
5. Review language and formatting issues.
6. Provide additional relevant references.

I hope you can complete the recommendation changes in a revision of your article.

Best,
Ana Maguitman

Reviewer 1 ·

Basic reporting

This paper is a pure application of deep neural networks and generally lie in the interest of researchers. This research makes use of machine deep ensemble model for providing high accuracy than several other models. This research work also provides a great comparison with other approaches. Overall, the look and idea of the paper is good but I have some suggestions to improve the quality and readability of the paper to develop more interest to the reader. Authors strictly follow my suggestions to meet the standard of prestigious journals like PeerJ Computer Science.
1. The abstract is too short and not covering all necessary parts. It should contain intro, problem, methodology, and results. It should be re-written
2. All learning models should be in a Table to look more compact and beautiful.
3. Results of table 3 and 4 should also be expressed in Figures. Specially comparison one.
4. I am not able to find a discussion that why the proposed model performs well and transformers model not?
5. Figure 2 caption should be self-explanatory.
6. Create a hyper-parameters table in which authors must add the configuration of all models to make these results reproduceable.
7. F1-Score, F1-scores, and F1 score should be sticked to one term.
8. Section 4.2, 4.5, and some other portion texts are out of canvas.
9. K-fold results are not same format like other result tables
10. Create a new subsection ‘Experimental Setup’ in the results section for showing the system configurations and language used for this research work.
11. I think authors need to adjust ELSTM-VC in the title to make it attractive.
12. English language editing’s and grammar check is needed.

Experimental design

Accurate

Validity of the findings

The experiments details all well documented.

Additional comments

Authors are advised to strictly follow the basic reporting comments.

Reviewer 2 ·

Basic reporting

no comment

Experimental design

no comment

Validity of the findings

no commet

Additional comments

This paper entitled ‘A novel approach for explicit song lyrics detection using machine and deep ensemble learning models’ is based on interesting idea as in this age of technology where the music is readily available to person of every age it is almost impossible to impose explicit content restrictions. The paper technically sounds very good because it is clear and understandable. It also provided numerous experiments to provide information about the proposed model that outperformed others. Results are presented in comprehensive manner and the subject of the paper is handled thoroughly and clearly.
It is recommended that authors supply more references about the applications of Deep Neural Network or ensemble models.
Formatting needs improvement as some portions of the text are out of canvas. The equations need to be in centre
Acronym table is necessary to understand the terms used in the paper.

Reviewer 3 ·

Basic reporting

This paper presents a natural language processing task with learning algorithms usage. The overall idea is good. The paper presentation is good. Experimentation is good. Still, this paper lacks some major edits to address before making any final decisions.

1. The results table formatting is not accurate throughout the paper. Authors should follow percentage accuracy with 2 decimal values precision.
2. Re-write the abstract to show the real problem and its solution.
3. Finally, English language editing’s necessary to make it more understandable.

Experimental design

1. There is no justification for why the authors used ETC and LSTM ensembles as well as LR and SGD ensembles.
2. I also want to see the results of RF and LSTM as RF and ETC have the same accuracy.
3. Why the comparison is done with a transformer-based model and not with SOTA models?

Validity of the findings

Further explanation is required why the author's ensemble only these models. Also want to see the results of other ensemble models

Additional comments

Overall the presentation and experimentation results are good but the research requires justification which is missing. The authors need to carefully support each result in the paper.

---

## Round 0.2 · accepted · Accept

Thank you for your contribution to PeerJ Computer Science and for addressing all the reviewers' suggestions. The reviewers are satisfied with the revised version of your manuscript and it is now ready to be accepted. Congratulations!

Reviewer 1 ·

Basic reporting

Authors carefully address all of my comments and the paper quality is significantly improved, therefore this paper is suitable for acceptance.

Experimental design

No comments

Validity of the findings

All of the required details are mentioned in the paper.

Additional comments

No further comments

Reviewer 3 ·

Basic reporting

No comments. Following my comments help a lot to improve this paper

Experimental design

No further comments.

Validity of the findings

Everything is clear now.

Additional comments

No further comments. The paper is improved pretty much and can be considered for publication in its current form.